# The Significance of the 98th Amino Acid in GP2a for Porcine Reproductive and Respiratory Syndrome Virus Adaptation in Marc-145 Cells

**DOI:** 10.3390/v16050711

**Published:** 2024-04-30

**Authors:** Yao Chen, Zhantang Huo, Qi Jiang, Zhiheng Qiu, Zheng Shao, Chunquan Ma, Guihong Zhang, Qi Li

**Affiliations:** 1School of Life Science and Engineering, Foshan University, Foshan 528011, China; yaochenvet@fosu.edu.cn (Y.C.); q1343384395@163.com (Z.H.); 15622057348@163.com (Z.Q.); chunquanma@163.com (C.M.); 2College of Veterinary Medicine, South China Agricultural University, Guangzhou 510520, China; jiangqi845@163.com (Q.J.); m17855760636@163.com (Z.S.); guihongzh@scau.edu.cn (G.Z.); 3MOA Key Laboratory of Animal Vaccine Development, Ministry of Agriculture, College of Veterinary Medicine, South China Agricultural University, Guangzhou 510520, China; 4College of Veterinary and National Engineering Research Center for Breeding Swine Industry, South China Agricultural University, Guangzhou 510520, China

**Keywords:** PRRSV, GP2a, adaptation, Marc-145 cells

## Abstract

Porcine reproductive and respiratory syndrome virus (PRRSV) is one of the most important pathogens in the pig industry. Marc-145 cells are widely used for PRRSV isolation, vaccine production, and investigations into virus biological characteristics. Despite their significance in PRRSV research, Marc-145 cells struggle to isolate specific strains of the North American virus genotype (PRRSV-2). The involvement of viral GP2a, GP2b, and GP3 in this phenomenon has been noted. However, the vital amino acids have not yet been identified. In this study, we increased the number of blind passages and successfully isolated two strains that were previously difficult to isolate with Marc-145 cells. Both strains carried an amino acid substitution in GP2a, specifically phenylalanine to leucine at the 98th amino acid position. Through a phylogenetic and epidemiologic analysis of 32 strains, those that were not amenable to isolation widely exhibited this mutation. Then, by using the PRRSV reverse genetics system, IFA, and Western blotting, we identified the mutation that could affect the tropism of PRRSV-2 for Marc-145 cells. Furthermore, an animal experiment was conducted. Through comparisons of clinical signs, mortality rates, and viral load in the organs and sera, we found that mutation did not affect the pathogenicity of PRRSV-2. In conclusion, our study firmly establishes the 98th amino acid in GP2a as a key determinant of PRRSV-2 tropism for Marc-145 cells.

## 1. Introduction

Porcine reproductive and respiratory syndrome (PRRS), caused by porcine reproductive and respiratory syndrome virus (PRRSV), is a highly devastating disease that profoundly affects the global swine industry [1,2]. The disease is characterized by severe reproductive failure in pregnant sows and persistent infection that results in respiratory tract distress [3]. PRRSV belongs to the family Arteriviridae and is classified into two species: Betaarterivirus suid 1 (PRRSV-1) and Betaarterivirus suid 2 (PRRSV-2) [4], which share approximately 60% genome sequence homology [5,6]. PRRSV has limited cell tropism; it infects only a few cell lines, such as primary porcine alveolar macrophages (PAMs), CL2621, blood monocytes, swine testis (ST) cells, and African green monkey kidney-derived cells, such as MA104 and Marc-145 cells [7,8,9,10]. PAMs are one of the target cells for PRRSV infection in vivo. They have a higher isolation rate than simian cell lines for primary isolation of PRRSV field strains. Nonetheless, PAMs have some disadvantages, including complicated procedures for manufacturing, a short lifespan, and ease of infection by other viruses. The other cell lines are not widely used in PRRSV research. Due to their low cost and ease of manipulation, Marc-145 cells have emerged as the most commonly used cells in PRRSV research, particularly in virus isolation, vaccine preparation, and investigation of PRRSV biological mechanisms [11,12,13]. In recent years, researchers first reported the existence of PRRSV-2 strains that could not be isolated using Marc-145 cells [14]. However, the mechanism has remained unclear.

In Arteriviruses, minor envelope proteins, including GP2, GP3, GP4, and E, play crucial roles in entry into cells [15]. In PRRSV-1, the amino acid 88/95 could improve viral growth in CL2621 cells [16], and the amino acid 88/94/95 in GP2a was responsible for adaptation to Marc-145 cells [17]. In PRRSV-2, researchers have demonstrated that GP2a, GP2b, and GP3 are involved in viral adaptation to Marc-145 cells through the substitution of large fragments [18], yet the vital amino acids have not been found.

This study aimed to find out which vital amino acid is the key determinant for PRRSV adaptation to Marc-145 cells. In addition, we wanted to research whether the key site could affect the PRRSV-2 replication ability, virulence, or immunogenicity.

## 2. Materials and Methods

### 2.1. Cell Culture and Viruses

Marc-145 cells and BHK-21 cells were maintained in Dulbecco’s modified Eagle’s medium (DMEM, Gibco, Gaithersburg, MD, USA) supplemented with 10% fetal bovine serum (FBS, Gibco, Gaithersburg, MD, USA). Primary porcine alveolar macrophages (PAMs) were cultured in RPMI 1640 medium (Gibco, USA). All cells were cultured in a 37 °C incubator with 5% CO_2_. PRRSV-2 strain XH-GD (GenBank accession no. EU624117) was preserved in our laboratory [19].

### 2.2. Virus Isolation and Plaque Purification

The positive clinical samples were homogenized and centrifuged, and the supernatants were filtered through a 0.22 μm filter and added to PAMs or Marc-145 cells to isolate the virus. The isolates were inoculated into Marc-145 cells or PAMs for 10 passages [20].

The virus was 10-fold serially diluted and cultured with Marc-145 cells. After 2 h of incubation at 37 °C, the plates were washed with sterile PBS three times. Then, the plates were overlaid with 2 mL 1.5% low melted agarose (Invitrogen, Waltham, MA, USA). Two days after infection, 2 mL 0.01% neutral red staining solution (Solarbio, Beijing, China) was added for staining at 37 °C for 3 h. A plaque was picked and inoculated into new Marc-145 cells. When the cells showed characteristic cytopathic effects (CPE), the cells were frozen and thawed twice. After centrifugation, the supernatants were collected, and two additional rounds of plaque purification were carried out [19]. All samples were stored at −80 °C.

### 2.3. Phylogenetic Analysis and Sequence Alignment

The full-length genomes of the isolates were amplified as described in previous reports [21], and the sequence was assembled by Lasergene 7.2 software and aligned with reference XH-GD strains by using MEGA software (version 6.0). A phylogenetic tree was generated with online tools (https://www.evolgenius.info/evolview, accessed on 14 June 2023) by the distance-based neighbor-joining method [22].

### 2.4. Full-Length PRRSV cDNA Clone Construction and Transfection

The pokXH-GD and pokLYNA plasmids were stored in our laboratory. The strategy for constructing the full-length cDNA clones of the pokXH-GD-98Phe and pokLYNA-98Phe strains is shown in Figure 1. Briefly, the amino acid residue at position 98 in GP2a from pokXH-GD and pokLYNA was mutated by a fusion PCR-based method, and the primers are listed in Table 1. The fully infectious clones were treated with BamH I and Kpn I. The mutant D fragments were then assembled into the backbones of the fully infectious clones. Two plasmids (pokXH-GD-98Phe and pokLYNA-98Phe) were generated. These plasmids were identified using DNA sequencing and transfected into BHK-21 cells as previously reported [21]. The viruses were recovered and evaluated with PCR and sequencing, and the viruses were named rXH-GD-98Phe and rLYNA-98Phe. All the viruses were stored at −80 °C.

### 2.5. Immunofluorescence Assay

PAMs or Marc-145 cells were grown on coverslips and then incubated with the virus at 37 °C for 1 h. The coverslips were washed with sterile PBS after infection and treated with paraformaldehyde for 30 min at 4 °C. Next, the fixed cells were permeabilized with 0.1% Triton X-100 for 30 min and blocked with 5% skim milk for 2 h. Then, the cells were incubated with mouse anti-N protein antibody (SOW17, Median, Seoul, Republic of Korea) (1:400 dilution) overnight at 4 °C. Subsequently, goat anti-mouse IgG Alexa Fluor 488 (Thermo, Waltham, MA, USA) (1:200 dilution) was used as a secondary antibody and incubated with the cells for 2 h. Finally, the nuclei were stained with 4,6-diamidino-2-phenylindole (DAPI, Solarbio, Beijing, China) for 5 min. Fluorescence was observed by fluorescence microscopy (ECHO, San Diego, CA, USA) [19].

### 2.6. Western Blotting

Marc-145 cells or PAMs were infected with PRRSV. At 48 h post-infection, the cells were treated with RIPA lysis buffer, and the samples were separated with SDS–PAGE (15%). Then, the proteins were transferred to PVDF membranes. Western blotting was performed as previously reported [19]. An anti-N protein antibody (SOW17, Median, Seoul, Republic of Korea) (1:1000 dilution) and horseradish peroxidase (HRP)-conjugated goat anti-mouse IgG (Beyotime, Nantong, China) were used as primary and secondary antibodies, respectively. Finally, the membranes were analyzed using Biosystems C280 (Azure, Dublin, CA, USA).

### 2.7. Quantitative Real-Time PCR

According to the instructions, a total RNA rapid extraction kit (Fastagen, Shanghai, China) was used to extract the total cellular RNA [23]. Then, 5 μg RNA of each sample was subsequently reverse transcribed to cDNA using a reverse transcription kit (Takara, Shiga, Japan). The cDNA was used as the template for the quantitative PCR assay using Premix Ex Taq™ (Takara, Japan) or TB Green^®^ Premix Ex Taq™ II (Takara, Japan) with a CFX96 real-time detection system (Bio-Rad, Hercules, CA, USA). The primers and probe sequences are listed in Table 2.

### 2.8. Multistep Growth Curve and Viral Titers

PAMs or Marc-145 cells were treated with the PRRSV at a multiplicity of infection (MOI) of 0.1. Then, the supernatants were collected at certain time points (8, 16, 24, 32, 40, and 48 h). Next, the viral titers, including the viral loads in infected PAMs, infected Marc-145 cells, serum, lungs, and lymph nodes, were calculated by quantitative PCR. For each round of quantitative PCR conducted with the probe method, a standard serially diluted XH-GD (10^0^–10^7^ TCID_50/mL_) was used to generate a standard curve (slope = −3.523; R^2^ = 0.993) [24].

### 2.9. Animal Experiment

Eighteen 4-week-old piglets (Landrace/Yorkshire) were obtained from a PRRSV-free farm; the average weight of the piglets was about 7.5 kg. All animals were confirmed to be negative for ASFV, PRV, SIV, CSFV, PCV2, and PRRSV using commercial real-time PCR kits and commercial ELISA kits.

According to the experimental design, the piglets were randomly divided into three groups (negative, rXH-GD, and rXH-GD-98Phe, n  =  6) arranged in different rooms at 26 °C. The piglets in the rXH-GD group and the rXH-GD-98Phe group were intranasally treated with 2 mL × 10^6^ TCID50 rXH-GD and rXH-GD-98Phe, respectively. The negative group received the same volume of DMEM via the same method as the placebo [24]. After viral infection, the clinical signs in every piglet were recorded every day, and serum samples were collected at 0, 3, 7, 10, and 14 days post-inoculation (dpi) to quantify the viral titer. The weight of each piglet was recorded every three days. At 7 dpi, one piglet from each group was randomly selected and euthanized by electric shock, and the lungs and lymph nodes were collected for quantifying the viral titer by quantitative PCR as previously reported [24]. All surviving piglets were euthanized at 14 dpi, and the lungs were collected for histological examination.

### 2.10. Statistical Analysis

Statistical analysis was carried out with GraphPad Prism (version 6.0). One-way analysis of variance (ANOVA) was used to analyze the differences among the groups. The differences in data were considered significant when *p* < 0.05.

## 3. Results

### 3.1. The LYNA and GDST Strains Do Not Easily Infect Marc-145 Cells

Two PRRSV-positive samples were collected from Henan Province and Guangdong Province, China. Reverse transcription PCR, sequencing, and phylogenetic analysis revealed that both viruses belonged to PRRSV-2. According to the collected place, the viruses were named LYNA and GDST, respectively. To isolate the viruses, we first used Marc-145 cells. After three blind passages, as previously reported [25,26], there were no significant cytopathic changes in Marc-145 cells, and the IFA showed similar results. These results suggested that LYNA and GDST exhibited poor adaptability to Marc-145 cells.

The adaptability of the virus to cells can be improved through continuous passaging [17,27,28]. Therefore, we revised the isolation methods and extended the number of passages from 3 to 10 in the Marc-145 cells. On the 10th passage, we observed typical cytopathic changes (Figure 2A), and the IFA results also confirmed that Marc-145 cells were infected with PRRSV (Figure 2B). This indicated that LYNA-3- and GDST-3 exhibited poor adaptability to Marc-145 cells. However, after 10 passages, GDST-P10 and LYNA-P10 were able to infect Marc-145 cells.

The quantitative PCR results revealed that the RNA copy numbers of LYNA and GDST remained low during generations P1–P5 of continuous passage. However, the copy number of LYNA increased significantly in passages P7–P8, while that of the GDST strain increased notably in generations P6–P7 (Figure 2C).

### 3.2. Sequencing Reveals That the 98th Amino Acid of GP2a May Be a Vital Site That Can Affect PRRSV Adaptation to Marc-145 Cells

After plaque purification, we obtained the viruses named LYNA-P10 and GDST-P10. Whole-genome sequencing was performed on the isolates LYNA, GDST, LYNA-P10, and GDST-P10 as previously reported [21]. Since the GP2a-GP3 region has been identified as a key factor related to the adaptability of PRRSV-2 to Marc-145 cells, we separately analyzed the GP2a-GP3 region. Compared with the original viruses, LYNA-P10 and GDST-P10 had only one mutation, in which the phenylalanine (Phe) at position 98 of GP2a was mutated to leucine (Leu) (Figure 3). Then, we collected 32 PRRSV-2 strains for which cell adaptation had been clearly identified, including 15 strains that could infect Marc-145 cells and 17 strains that could not (Table 3) [18,29,30]. By analyzing the GP2a genes of those viruses, we found that strains that had difficulty infecting Marc-145 cells existed in lineage 8, lineage 3, and lineage 1, and all strains that had difficulty infecting Marc-145 cells had the 98Phe mutation of GP2a (Figure 4). This result indicated that the mutation may affect the ability of PRRSV-2 to adapt to Marc-145 cells.

### 3.3. Identification of an Amino Acid Mutation at the 98th Position That Can Change the Cellular Tropism of PRRSV-2 towards Marc-145 Cells

To investigate whether the 98th amino acid of GP2a could alter the adaptation of PRRSV-2 to Marc-145 cells, we constructed four fully infectious clones: pokXH-GD, pokXH-GD-98Phe, pokLYNA-P10, and pokLYNA-P10-98Phe. The sequences of all the plasmids were confirmed by sequencing, and then the plasmids were transfected into BHK-21 cells to rescue the virus. After 48 h, the supernatants were collected and inoculated into PAMs. After three passages, the infected PAMs were lysed with RIPA buffer for Western blotting. The results showed that the mutation did not affect virus rescue, and all viruses were able to replicate in PAMs (Figure 5A). However, when Marc-145 cells were incubated with BHK-21 supernatant, only the parent viruses (rXH-GD and rLYNA-P10) exhibited specific fluorescence or typical cytopathic effects (CPEs), while the two mutated viruses (rXH-GD-98Phe and rLYNA-P10-98Phe) did not show specific fluorescence or significant CPEs (Figure 5B,C). Those results indicated that the mutated viruses were unable to replicate in Marc-145 cells. These results suggested that a mutation in position 98 of GP2a affects the ability of PRRSV-2 to adapt to Marc-145 cells. The qPCR results also prove the same conclusion.

### 3.4. The Mutation Does Not Impact the Pathogenicity of PRRSV-2

The replication ability reported previously may affect the pathogenicity of PRRSV [24]. In PRRSV-1, amino acid mutations in GP2a have been found to enhance viral growth in Marc-145 cells [16,17]. Thus, we investigated whether the mutated viruses could influence PRRSV-2 replication in PAMs or virulence in piglets. We compared the replication ability between the mutant strain and parent strains using growth curves. The results demonstrated that the mutation did not reduce the replication ability of PRRSV-2 in PAMs (Figure 6).

To investigate the impact of the mutation on viral pathogenicity, animal experiments were conducted. Compared to the negative controls, the animals in the rXH-GD and rXH-GD-98Phe groups exhibited typical clinical symptoms of PRRSV infection, such as inappetence, lethargy, dyspnea, periocular and eyelid oedema, and hyperspasmia. As shown in Figure 7A, the average temperature in the rXH-GD group and the rXH-GD-98Phe group exceeded 40.5 °C from days 7 to 14 post-infection (dpi). There was no significant difference in mean body temperature between the rXH-GD and rXH-GD-98Phe groups at any time point. We calculated the average weight in each group and observed that the body weight in the negative group continued to increase, whereas the body weights in the rXH-GD group and rXH-GD-98Phe group increased only slightly (Figure 7B). The body weights in the two infected groups did not differ significantly at any time point but were significantly lower than that in the negative control group at 6–12 dpi. Regarding mortality, the mortality in the rXH-GD group reached 50%, with piglets dying at 8, 8, and 13 dpi. The rXH-GD-98Phe group also had a 50% mortality rate, with three piglets dying at 7, 10, and 12 dpi (Figure 7C).

We collected lungs to analyze the histopathological sections and macroscopic lesions at 14 dpi. The viral load was measured by quantitative PCR as previously reported. For each round of quantitative PCR by the probe method, a standard serially diluted PRRSV strain XH-GD (100–107 TCID50/mL) was used to generate a standard curve (slope = −3.523; R^2^ = 0.993).

The alveolar interstitial spaces in the lungs of both the rXH-GD group and the rXH-GD-98Phe group were wider than those of the negative control group. Inflammatory cells accumulated in the interlobular septal and bronchus were also found in the rXH-GD group and the rXH-GD-98Phe group. Those results suggested that the lungs in the rXH-GD group exhibited lesions and histopathological changes similar to those observed in the rXH-GD-98Phe group, as illustrated in Figure 8. Overall, the mutant virus demonstrated comparable clinical pathogenicity in piglets when compared to the parental strain. To summarize, piglets in both the rXH-GD and rXH-GD-98Phe groups exhibited heightened rectal temperatures and severe clinical symptoms, including reduced appetite, lethargy, difficulty in breathing, swelling around the eyes and eyelids, and increased muscle spasms, when compared to negative control subjects.

### 3.5. The Mutation Does Not Impact the Replication Ability of PRRSV-2 In Vivo

To further investigate the impact of the mutation on viral replication and immunogenicity in vivo, we assessed viral loads in sera, lungs, and lymph nodes. Serum samples were collected from each piglet at 3, 7, 10, and 14 dpi. At 7 dpi and 14 dpi, a piglet was randomly picked and euthanized, and its lungs and lymph nodes were collected. Viral copy numbers in the lymph nodes, lungs, and serum were calculated using RT–qPCR. As shown in Figure 9A, in the serum, the viral load peaked at 10 dpi, and there was no significant difference in the viral load between the two infected groups (Figure 9A). In the lungs, both groups showed a similar trend in viral load, which reached its peak at 14 dpi (Figure 9B). Nevertheless, the viral load of rXH-GD-98Phe in the lymph nodes was noticeably higher (in 7 dpi, *p* = 0.0013, in 14 dpi, *p* = 0.0084) than that of rXH-GD, raising questions about whether this mutation affects the replication or survival capabilities of PRRSV in the lymph nodes (Figure 9C). Considering all viral load results, we concluded that the replication efficiency of rXH-GD-98Phe in vivo was similar to that of the parental virus.

## 4. Discussion

Marc-145 cells are among the few types of cells that can be infected by PRRSV and play an important role in PRRSV research, especially in PRRSV-2 isolation, vaccine production, and the study of PRRSV-2 biological characteristics [31,32]. Researchers have increasingly observed that certain PRRSV-2 strains are unable to infect Marc-145 cells [14], thereby restricting the ability to study virus–host interactions, characterize gene function, and produce vaccines on a large scale.

In this study, we revised the isolation method and successfully obtained the LYNA and GDST strains by extending the passage time in Marc-145 cells. In fact, we used eight PRRSV-2 strains, which were not easily isolate by using the traditional method in this experiment. The isolation rate was only 25 percent by extending the passage time. This indicates that the PAMs are the better choice to isolate the special PRRSV-2.

GP2a is a minor structural glycoprotein of PRRSV, it has been proven that it could affect the viral cell adaptation. In PRRSV-1, the amino acid 88/94/95 of GP2a have been found to improve viral adaptation to CL2621 cells or Marc-145 cells [16,17]. Regarding the mechanism of PRRSV-2 cell adaptation changes, previous studies have demonstrated that variations in the GP2a-GP3 complex were the primary determinants of PRRSV-2 tropism in Marc-145 cell culture [18]. The K160 residue in GP2 is one of the vital determinants of PRRSV tropism to PAMs [33]. In our study, we compared LYNA and GDST with LYNA-P10 and GDST-P10. Through sequencing, we identified a mutation, phenylalanine to leucine, at the 98th amino acid of the GP2a protein among the isolates. Subsequently, we collected 32 strains, including 15 strains capable of infecting Marc-145 cells and 17 strains that had difficulty infecting Marc-145 cells. The same mutations were also found in the PRRSV strains that had difficulty infecting Marc-145 cells. This indicates that the 98th amino acid of GP2a plays a crucial role in PRRSV adaptation to Marc-145 cells. Finally, we used LYNA-P10 and GDST-P10 as parental strains to construct two infectious clones via a reverse genetic manipulation platform. We introduced a mutation at the 98th amino acid of GP2a. Through IFA and Western blotting, we discovered that the mutation caused PRRSV-2 to be incapable of infecting Marc-145 cells. These results suggested that the 98th amino acid of GP2a in PRRSV-2 impacts PRRSV-2 infectivity in Marc-145 cells.

The GP2a is a structural protein; it could interact with the other viral proteins and attaches to the receptor proteins for virus entry into cells [34,35]. Since mutation of one site in GP2a did not affect the rescue of PRRSV-2 from BHK-21 cells and all mutated PRRSV-2 could replicate in PAMs as the parental virus, we guessed the mutation did not affect the replication and assemblage of PRRSV-2. In the meantime, considering the function of GP2a, we speculate that the mutation may affect PRRSV-2 uncoating or attachment in Marc-145 cells. This means that the mutation of position 98 of GP2a affects the interaction between the GP2a and Marc-145 cells receptor proteins, making it difficult for PRRSV-2 to infect Marc-145 cells.

Previous studies found that the CD163 was considered as the primary and core receptor for PRRSV and determines the susceptibility of cells to the PRRSV [36]. Previous reports found when Marc-145 cells overexpress the porcine CD163 protein, the PRRSV-2 isolation rate would significantly increase [37]. It is indicated that there is a difference between monkey CD163 and porcine CD163. GP2a could interact with CD163-SRCR5 domain [34], so we suspected that the mutation could affect the GP2a to interact with monkey CD163-SRCR5 domain. However, the mutation may affect the PRRSV-2 to binding with other cellular receptors; the exact underlying mechanism remains unclear and requires further investigation.

In the diagnosis of PRRS in veterinary practice, the suspected positive samples are usually first tested with conventional PCR. If the result is positive, the samples will be treated and inoculated onto cells for isolating the virus. We advise researchers to analyze the GP2a sequence before attempting PRRSV isolation. The analysis result could guide the researchers to select the appropriate cells to isolate PRRSV-2. It will improve the efficiency of PRRSV-2 isolation.

In this study, we also investigated the effect of this mutation on PRRSV infection and pathogenicity in piglets. We inoculated piglets with rXH-GD-98Phe and rXH-GD. The animal experiment showed that the mutant strain and the parental virus exhibited similar virulence, lesion appearance, and pathology results. These findings indicate that the mutation does not affect PRRSV-2 virulence.

In summary, our findings suggest that increasing the number of blind passages may be a viable strategy for isolating PRRSV strains that are difficult to isolate in Marc-145 cells. Furthermore, the mutation of phenylalanine to leucine at the 98th amino acid in GP2a can alter PRRSV adaptation to Marc-145 cells, making this amino acid site a potential marker for assessing the suitability of PRRSV-2 strains for isolation in Marc-145 cells. However, the underlying mechanism by which this mutation affects PRRSV fitness remains to be investigated.

## Figures and Tables

**Figure 1 viruses-16-00711-f001:**
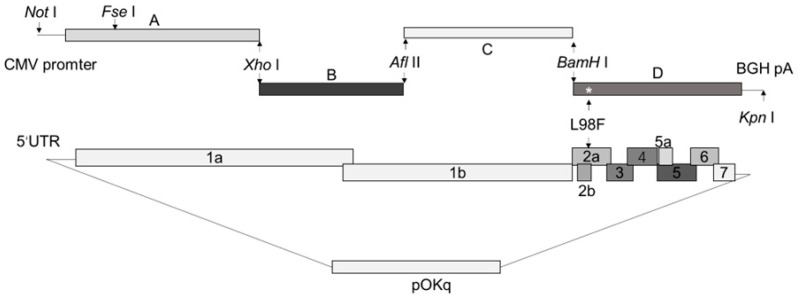
Strategy used for constructing the full-length PRRSV strain.

**Figure 2 viruses-16-00711-f002:**
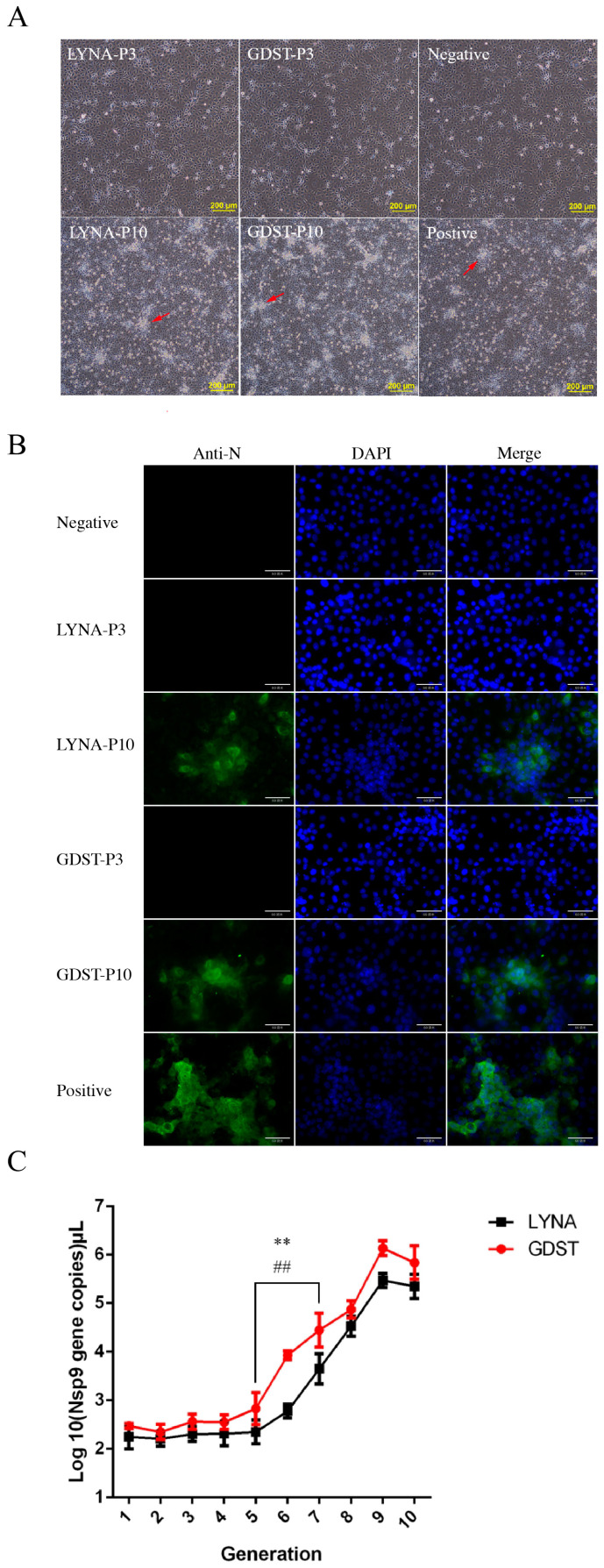
Isolation of LYNA and GDST PRRSV-2 viruses. (**A**) Cytopathic effects of virus infection for different durations in Marc-145 cells; red arrow indicates the typical cytopathic changes. (**B**) IFA results for the different durations of virus infection. (The size of the scale bar is 60 μm). (**C**) Viral RNA copy numbers of LYNA and GDST during continuous passage in Marc-145 cells (*: LYNA; ^#^ GDTS). The data shown represent the mean ± SD (*n* = 3). (**,^##^ *p* < 0.01).

**Figure 3 viruses-16-00711-f003:**
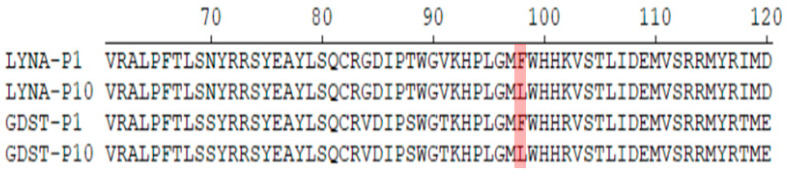
The 98th amino acid of GP2a was mutated during passaging. The color indicated the 98th amino acid of GP2a.

**Figure 4 viruses-16-00711-f004:**
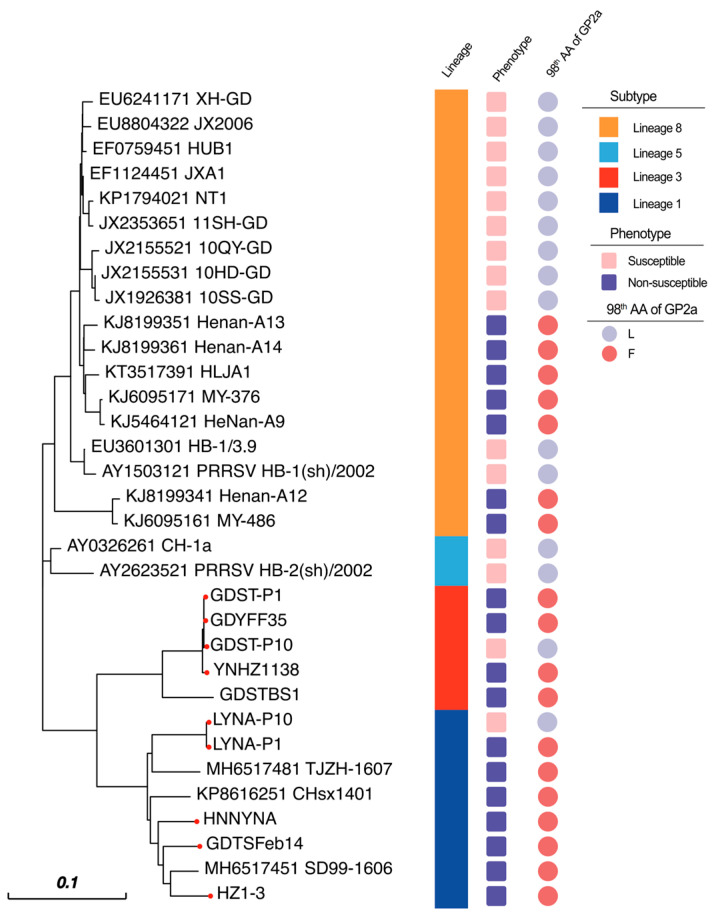
Phylogenetic and epidemiological analyses of several common subtypes of PRRSV-2 strains that could infect or not infect Marc-145 cells. There may be a certain correlation between whether a strain can infect Marc-145 cells and whether the 98th amino acid in the GP2a gene is leucine. (The red dots indicate the strains stored in our laboratory).

**Figure 5 viruses-16-00711-f005:**
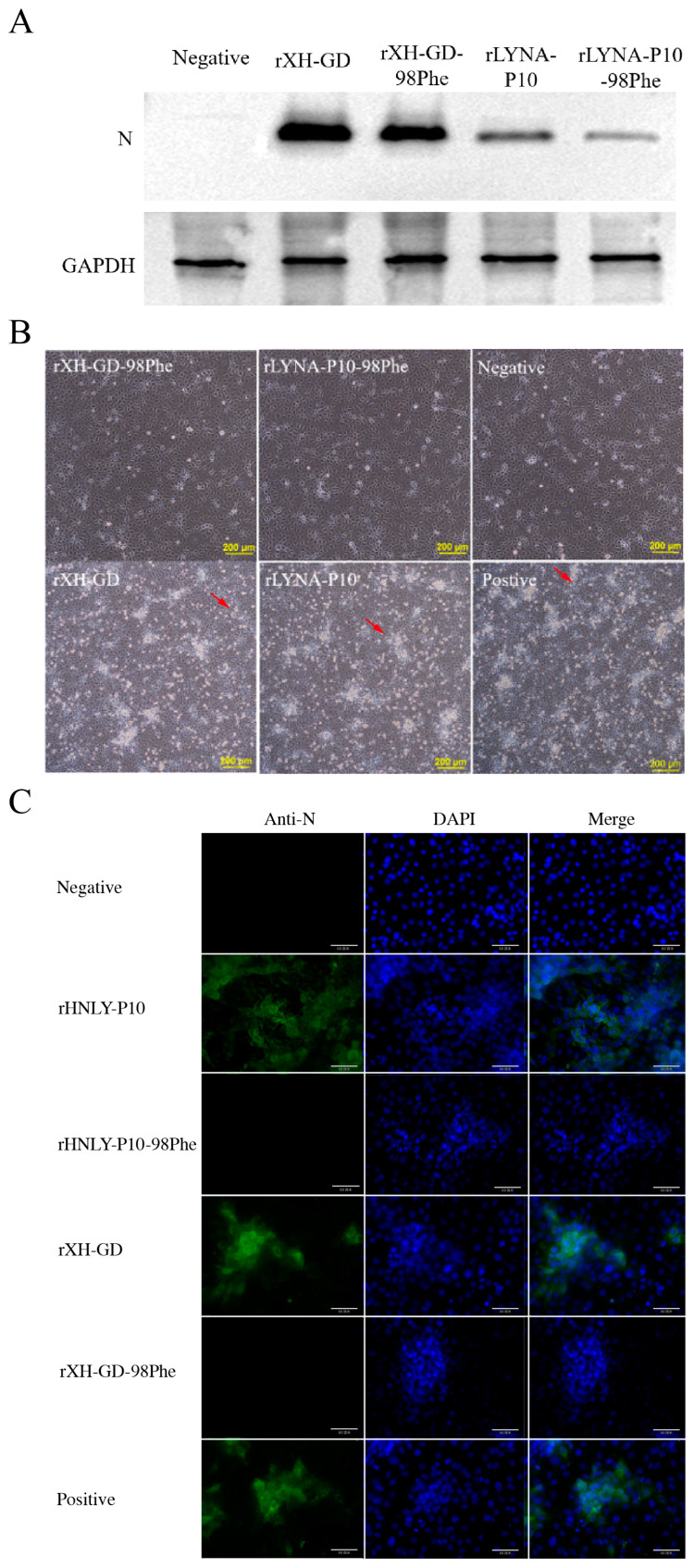
The mutation of GP2a indeed changed the cellular tropism of PRRSV-2. (**A**) The mutation did not affect virus rescue. (**B**) Cytopathic effects of the parent viruses and the mutated viruses in Marc-145 cells; red arrow indicates the typical cytopathic changes. (**C**) IFA results. Marc-145 cells were infected with rXH-GD, rLYNA-P10, rXH-GD-98Phe, and rLYNA-P10-98Phe. The infected cells were fixed with PBS and incubated with mouse anti−N mAb followed by FITC goat anti-mouse IgG (H + L). The nuclei were stained with DAPI. (The size of the scale bar is 60 μm).

**Figure 6 viruses-16-00711-f006:**
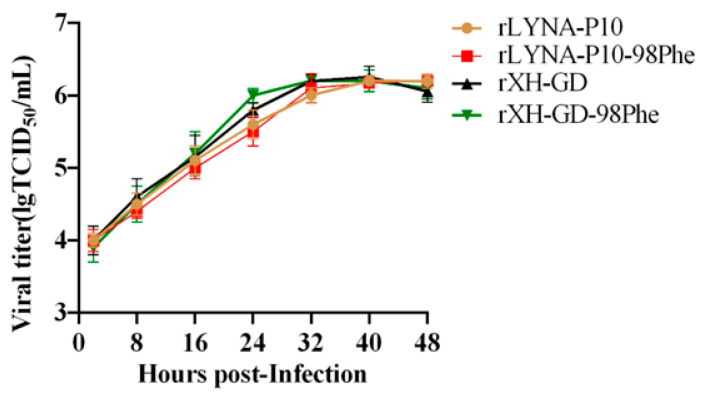
The mutation did not impact the PRRSV-2 replication ability in PAMs. The data shown represent the mean ± SD (*n* = 3).

**Figure 7 viruses-16-00711-f007:**
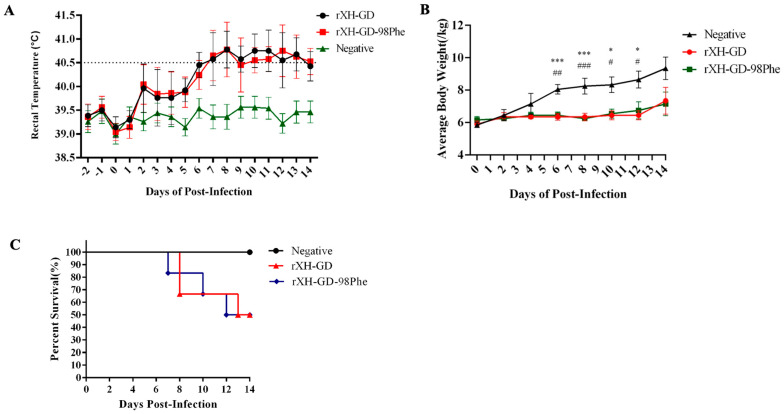
Mutation of the 98th residue of GP2a did not affect the pathogenicity of PRRSV-2. (**A**) Daily rectal temperatures of the pigs in the rXH-GD, rXH-GD-98Phe, and mock groups. (**B**) Body weight gain in different groups during the experiment. (*,^#^ *p* < 0.05; ^##^ *p* < 0.01; ***,^###^ *p* < 0.001) (**C**) The mortality of each group was recorded daily until 14 dpi and used to calculate the survival rate. The values are presented as the mean ± standard deviation (shown with error bars) (*: rXH-GD vs. Negative; ^#^ rXH-GD-98Phe vs. Negative).

**Figure 8 viruses-16-00711-f008:**
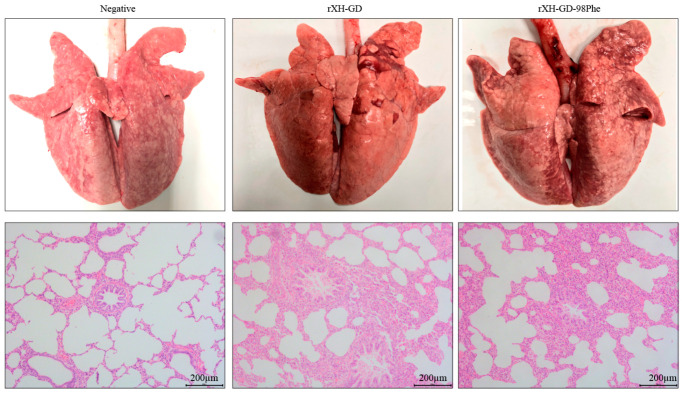
Macroscopic lung lesions and microscopic lung lesions of piglets. Severe lung lesions in the rXH-GD group and rXH-GD-98Phe group were characterized by swelling, congestion, fibrosis, and inflammatory cell aggregates.

**Figure 9 viruses-16-00711-f009:**
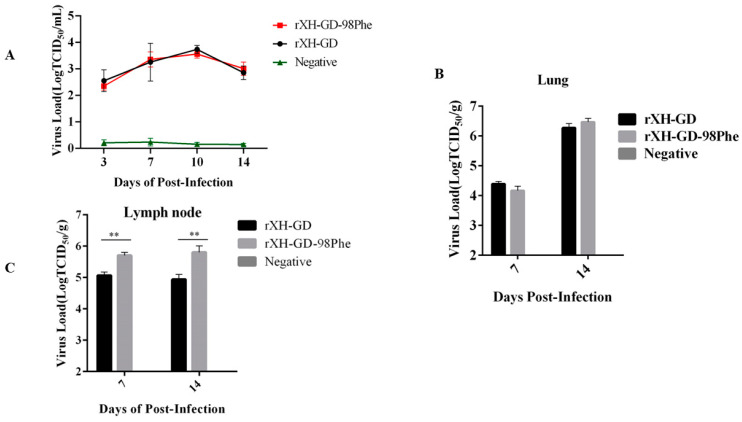
Viral loads in vivo in the different groups. (**A**) Concentration of PRRSV gRNA in the serum per milliliter. (**B**) Expression of PRRSV gRNA in lymph nodes per gram. (**C**) Level of PRRSV gRNA in the lungs per gram. The data shown represent the mean ± SD (*n* = 3) (** *p* < 0.01).

**Table 1 viruses-16-00711-t001:** The primers for PRRSV infectious clone construction.

Name	Primer Sequence (5′-3′)	Product Size (bp)
XH-D-F	CAACTTGAAGGCCGCCATTTTACCT	783
XH-98Phe-R	GACACCTTATGGTGCCAAAACACCCCCAAAG	
XH-98Phe-F	GCGTCAACACCCTTTGGGGGTGTTTTGG	3455
XH-D-R	GCGAATTGGGGATCGAGGTACCCAGAA	
LYNA-D-F	GCTTACCTCCATCCAGGG	875
NA-98Phe-R	TATGGTGCCAAAACATCCCCAAG	
NA-98Phe-F	CCCTTGGGGATGTTTTGGCACCATAGG	3519
LYNA-D-R	GGCGATTAAGTTGGGTAACG	
LYNA-A-F	TACACCGGTAACGTCATGACGTATAGGTGTTGGC	2185
LYNA-A-R	AACAACCACTCCAACTCCAG	
LYNA-B-F	CCTCCGCGGTGCAGCAAGTCCTGAAG	5130

**Table 2 viruses-16-00711-t002:** The primers for real-time PCR.

Name	Primer Sequence (5′-3′)	Product Size (bp)
RTNsp9-F	CCCTCCATGCCAAACTACCAC	194
RTNsp9-R	TTGTCTTCTTTGGGTCCGTCT	
GAPDH-F	CTGCCGCCTGGAGAAACCT	250
GAPDH-R	GCTGTAGCCAAATTCATTGTCG	
qNsp9-F	CCTGCAATTGTCCGCTGGTTTG	146
qNsp9-R	GACGACAGGCCACCTCTCTTAG	
Probe-Nsp9	FAM-ACTGCTGCCACGATTTACTGGTCACGCAGT-BHQ1	

**Table 3 viruses-16-00711-t003:** The list of strains.

	Virus Name	GenBank Accession No.	98th Animo Acid of GP2a	Able to Adapt to Marc-145 Cells
1	XH-GD	EU624117.1	L ^a^	Y ^c^
2	JX2006	EU880432.2	L	Y
3	HUB1	EF075945.1	L	Y
4	JXA1	EF112445.1	L	Y
5	NT1	KP179402.1	L	Y
6	11SH-GD	JX235365.1	L	Y
7	10QY-GD	JX215552.1	L	Y
8	10HD-GD	JX215553.1	L	Y
9	10SS-GD	JX192638.1	L	Y
10	Henan-A13	KJ819935.1	F ^b^	N ^d^
11	Henan-A14	KJ819936.1	F	N
12	HLJA1	KT351739.1	F	N
13	MY-376	KJ609517.1	F	N
14	HeNan-A9	KJ546412.1	F	N
15	HB-1/3.9	EU360130.1	L	Y
16	HB-1(sh)/2002	AY150312.1	L	Y
17	Henan-A12	KJ819934.1	F	N
18	MY-486	KJ609516.1	F	N
19	CH-1a	AY032626.1	L	Y
20	HB-2(sh)/2002	AY262352.1	L	Y
21	GDST-P1	-	F	N
22	GDSTF35	-	F	N
23	GDST-P10	-	L	Y
23	YNHZ1138	-	F	N
24	GDSTBS1	-	F	N
25	LYNA-P10	-	L	Y
26	LYNA-P1	-	F	N
27	TJZH-1607	MH651748.1	F	N
28	CHsx1401	KP861625.1	F	N
29	LYNA-N	-	F	N
30	GDTSFeb14-N	-	F	N
31	SD99-1606-N	MH651745.1	F	N
32	HZ1-3	-	F	N

^a^ L: Leu; ^b^ F: Phe; ^c^ Y: Yes; ^d^ N: No.

## Data Availability

Data are available upon request from the corresponding author.

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
