# Peer review of "The Significance of the 98th Amino Acid in GP2a for Porcine Reproductive and Respiratory Syndrome Virus Adaptation in Marc-145 Cells"

_viruses, 2024, doi:10.3390/v16050711_

Round 1
Reviewer 1 Report
Comments and Suggestions for Authors
The authors describe the importance of the 98th Amino acid in GP2a for porcine reproductive and respiratory syndrome virus adaption in Marc-145 cells.
General comment: The authors use significance, significantly or significant several times in the article. Can you please clarify what you mean by this statement or add a p-value to it?
General: Please add indication of how you named “LYNA” and “GDST” as this is not clear.
Materials and Methods, cell culture and viruses: Mention how other virus isolates were obtained.
Materials and Methods, line 130: Check English for use of “served used”.
Materials and Methods, line 151: “All test results were negative”. Please remove here and see if discussed in results section.
Figure 2: Add as follows “Isolation of LYNA and GDST PRRSV-2 viruses”.
Figure 3: Change “color was” to “Color indicates the”.
Figure 4: Mention other strains sequences obtained from GenBank.
Table 3: Change “Whether to adapt March-145 cells” to “Able to adapt to Marc-145 cells”. Add Note below Table to explain abbreviations L, F, Y and N.
Results, line 234: Add “(not shown)” after sentence on qPCR results.
Results, line 298: Re-word as methodology describes this better. It is unclear from this that they were different pigs on different days.
Results, line 304: Add p-value after “noticeably higher”.
Results, line 309-311: Was this a comment from reviewer or editor as it does not fit in text and I suggest deleting it.
Discussion, line 326-329: Re-word sentence as it is confusing.
Discussion, line 331: Change “proved that could” to “proven that it could”.
Discussion, line 346: Please explain how Western blot shows that PRRSV-2 is incapable of infecting Marc-145 cells. It indicates protein presence or absence not infection, however, the latter has an influence on infection.
Discussion, line 350: Change “is structural protein” to “is a structural protein”.
Discussion, line 351: Change “mutated” to “mutation of”.
Discussion, line 352-353: Re-word to clarify sentence.
Discussion, line 353-354: Check sentence punctuation.
Discussion, line 358: You did not describe use of “CD163” before thus add indication that this was in previous study or another study.
Discussion, line 361-363: Please use of English with focus on “different”, “So” and “mutated”.
Discussion, line 367: Usually we refer to “conventional” rather than “classic” PCR.
Comments on the Quality of English Language
Please check English language to be appropriate and understandable. See some comments in review, but additional English assistance may be required.
Reviewer 2 Report
Comments and Suggestions for Authors
This manuscript demonstrates that mutation in amino acid 98 of GP2 of PRRSV (L by F) is responsible for the low adaptation of the virus to replicate in MARC-145 cells. The authors performed an interesting and complete study to support the conclusion. I have some suggestions to improve the manuscript.
I suggest using F instead of "Phe" throughout the manuscript.
Use 98 instead of 98th, to describe the amino acid position throughout the manuscript.
The quality of the figures describing the cytopathic effect is poor; please improve.
The mutation does not impact the replication ability or immunogenicity of PRRSV-2 in vivo, change to "The mutation does not impact the replication ability of PRRSV-2 in vivo". The authors did not evaluate the immunogenicity. Line 295
Check the spacing between the dot and references and caps after the dot.
Lines 358-365 are irrelevant; remove them.
Please remove the following: "This section may be divided by subheadings. It should provide a concise and precise description of the experimental results, their interpretation, and the experimental conclusions that can be drawn."
Comments on the Quality of English LanguageNone
